# Performance of Winter Wheat Cultivars Grown Organically and Conventionally with Focus on Fusarium Head Blight and *Fusarium* Trichothecene Toxins

**DOI:** 10.3390/microorganisms7100439

**Published:** 2019-10-11

**Authors:** Tomasz Góral, Aleksander Łukanowski, Elżbieta Małuszyńska, Kinga Stuper-Szablewska, Maciej Buśko, Juliusz Perkowski

**Affiliations:** 1Department of Plant Pathology, Plant Breeding and Acclimatization Institute—National Research Institute, Radzików, 05-870 Błonie, Poland; 2Department of Phytopathology and Molecular Mycology, Faculty of Agriculture and Biotechnology, UTP University of Science and Technology, Al. prof. S. Kaliskiego 7, bldg. I, 85-796 Bydgoszcz, Poland; luk-al@utp.edu.pl; 3Department of Seed Science and Technology, Plant Breeding and Acclimatization Institute—National Research Institute, Radzików, 05-870 Błonie, Poland; e.maluszynska@ihar.edu.pl; 4Department of Chemistry, Poznań University of Life Sciences, ul. Wojska Polskiego 75, 60-625 Poznań, Poland; kinga.stuper@up.poznan.pl (K.S.-S.); maciej.busko@up.poznan.pl (M.B.);

**Keywords:** Fusarium head blight, *Fusarium* species, soil minerals, mycotoxins, organic farming, sowing value, winter wheat

## Abstract

Growing acreage and changing consumer preferences cause increasing interest in the cereal products originating from organic farming. Lack of results of objective test, however, does not allow drawing conclusions about the effects of cultivation in the organic system and comparison to currently preferred conventional system. Field experiment was conducted in organic and conventional fields. Thirty modern cultivars of winter wheat were sown. They were characterized for disease infection including Fusarium head blight, seed sowing value, the amount of DNA of the six species of *Fusarium* fungi as well as concentration of ergosterol and trichothecenes in grain. The intensity Fusarium head blight was at a similar level in both systems. However, *Fusarium* colonization of kernels expressed as ergosterol level or DNA concentration was higher for the organic system. It did not reflect in an increased accumulation of trichothecenes in grain, which was similar in both systems, but sowing value of organically produced seeds was lower. Significant differences between analyzed cropping systems and experimental variants were found. The selection of the individual cultivars for organic growing in terms of resistance to diseases and contamination of grain with *Fusarium* toxins was possible. Effects of organic growing differ significantly from the conventional and grain obtained such way can be recommended to consumers. There are indications for use of particular cultivars bred for conventional agriculture in the case of organic farming, and the growing organic decreases plant stress resulting from intense fertilization and chemical plant protection.

## 1. Introduction

A way of growing crops is changing because of the geopolitical situation and consumer preferences. In recent years, high interest in organic farming has been observed in Europe (https://ec.europa.eu/eurostat/statistics-explained/pdfscache/5461.pdf). In 2002, organic farming took up 5.0 million hectares, while in 2017 it was 12.6 million hectares. Austria, Estonia, Sweden, Italy, Czech Republic, and Latvia were the countries with the highest share of organic farmland, while the largest areas of organic farmland were in Spain, Italy, France, and Germany. In Poland in 2016 it was 536.6 thousand ha (3.7% of all agricultural land) [1].

This is due to the awareness that in organic farming practices the use of artificial fertilizers as well as pesticides is not allowed. There is limited list of substances, which can be used as natural fungicides to protect crops against fungal diseases. Lack of fungicide protection can result in higher severity of fungal diseases. Chemical seed treatment is not applied which leads to increased incidence of seed borne diseases [2,3]. Thus, seed transmitted diseases are considered the most harmful in organic farming. Leaf diseases (not seed transmitted) and foot rots are less important. Severity of these diseases correlates with high nitrogen doses and high crop density, so under organic farming conditions they are less damaging [4]. Diseases caused by fungi surviving on crop debris (including Fusarium head blight) can be controlled by cultural practices, so they are less damaging than seed borne ones [3]. However, *Fusarium* fungi causing Fusarium head blight are able to produce toxic secondary metabolites–mycotoxins contaminating grain. The main *Fusarium* species causing Fusarium head blight are *F. culmorum*, *F. graminearum,* and *F. avenaceum* [5,6]. Cereal heads are infected mainly during the flowering period [7]. This is the stage where cereals are the most susceptible to infection with *Fusarium* fungi spores. After infection, the fungus develops in infected flower spreading then to other flowers in the spikelet. Then through rachis, the fungus spreads to another spikelet causing necrosis and bleaching individual spikelets [8,9]. The invaded cereal grain, even visually healthy looking, is contaminated with fungal mycotoxins, which are phyto- and zootoxic. *Fusarium* spp. affecting cereals are known as potent producers of type A trichothecenes (T-2 and HT-2 toxins, diacetoxyscirpenol et al.) and of type B (deoxynivalenol, nivalenol et al.) as well as moniliformin, zearalenone, enniatins, beauvericin, and the other toxins [5,6,10].

Avoiding the presence of *Fusarium* mycotoxins in food is very important, thus organic food is perceived as “food without chemistry” of higher quality than conventional [11,12]. In the literature, you can find analyses on this issue comparing wheat form organic and conventional cropping systems [13,14,15,16,17]. Mäder et al. [18] analyzed *Fusarium* metabolites, deoxynivalenol (DON) and nivalenol (NIV), content in wheat grain produced in a 21-year conventional and organic agrosystems. It was found higher concentration of DON in samples from conventional fields in both years of mycotoxin analysis; however, differences were not significant. NIV concentration was similar in both cropping systems. Magkos et al. [19] in their review summarized results of 12 papers on contamination of organic and conventional cereals with *Fusarium* mycotoxins. Organically grown cereals has been reported to be either more, less, or equally contaminated compared with conventional cereals. Authors concluded that this variability resulted from different cultivars, geographical locations of fields and time of harvest in different studies. It makes data not directly comparable.

In the literature, it can be found a number of analyses of effects of organic cultivation of wheat. However, experimental data that can verify the views presented there are still not very numerous. Considering this, we decided to carry out a field experiment on sowing 30 cultivars of winter wheat in the same location, at the same time on conventional and organic plots. The aim of the experiment was a comprehensive comparison of the results obtained for both cropping systems through the analysis in a series of elements that describe the structure of the yield, fungal diseases, presence of *Fusarium* fungi through analysis of the DNA content, production of mycotoxins in grain. The results were subject of the widest possible statistical analysis with the aim of finding relevant or irrelevant differences in both cultivation systems.

## 2. Material and Methods

### 2.1. Field Experiments

Thirty cultivars of winter wheat (*Triticum aestivum* L.) were evaluated (Table 1). The cultivars were listed in the Polish National List of the Research Centre for Cultivar Testing (COBORU) and were added to the list between 1998 (‘Mewa’) and 2009 (‘Belenus’). The cultivars were described in detail in the paper of Góral et al. [20]. They differed in the pedigree, morphological characters, and resistance to Fusarium head blight (FHB). Cultivars were grouped in four classes of FHB resistance: susceptible (S), medium susceptible (MS), medium resistant (MR), and resistant (R).

Field experiments were established in 2014 in the experimental fields of state-owned research institute—Plant Breeding and Acclimatization Institute (IHAR-PIB) in Radzików, Central Poland. First experiment was sown in the conventional field (GPS coordinates: 52.212517, 20.634765). Pre-crop was oilseed rape. Artificial fertilizers were applied according to standard agricultural practices in IHAR-PIB in particular. In the autumn 3 dt ha^−1^ of ‘Polifoska 6′ fertilizer was applied (N—18 kg ha^−1^, P—45 kg ha^−1^, K—72 kg ha^−1^). In the spring, after the start of vegetation ammonium nitrate fertilizer was applied in an amount providing 68 kg N ha^−1^. Weeds and pests were controlled with herbicides and insecticides. Immediately after sowing weeds were controlled with herbicide ‘Maraton 375SC’ in a dose of 4 L ha^−1^. In spring weeds were controlled using the herbicide ‘Attribut 70GS’ in a dose of 60 mg ha^−1^. Cereal leaf beetle and aphids were controlled with ‘Fastac Active 050ME’ in a dose of 250 mL ha^−1^. No fungicides were applied.

Simultaneously the same wheat cultivars were sown in the experimental organic field of IHAR-PIB (GPS coordinates: 52.216319, 20.638653). Wheat was grown according to organic farming practices with no chemical disease control and application of fertilizers. Pre-crop was pea. Weeds were controlled mechanically. No fertilizers or other components allowed in organic farming were applied. Distance between two experimental fields was about 500 m. Single plot size in both experiments was 5 m^2^. In both fields, cultivars were sown in three randomized blocks (replications) distant from each other by 2 m.

Heading and full flowering dates for individual plots were recorded. Plant height was measured after the end of heading stage. Fusarium head blight was scored based on the mean percentage of blighted spikelets per infected head (disease severity) and the percentage of infected heads per plot (disease incidence). Fusarium head blight index (FHBi) was calculated as the combination of disease severity and disease incidence.
(1)$${\mathrm{FHB}}_{i}=\left({\mathrm{FHB}}_{\mathrm{severity}}\times {\mathrm{FHB}}_{\mathrm{incidence}}\right)/100$$

Presence of other fungal diseases were also recorded. They were as follows: yellow rust (*Puccinia striiformis*), leaf rust (*P. triticina*), Septoria tritici blotch (*Zymoseptoria tritici*), Stagonospora nodorum blotch (*Parastagonospora nodorum*) and tan spot (*Pyrenophora tritici-repentis*). These diseases were scored according to percentage of leaf area per plot with symptoms of disease—necrosis and/or sporulation.

### 2.2. Analysis of Mineral Elements in Soil

In spring, soil samples were collected from conventional and organic fields. Twenty soil cores were taken from experimental plots in both fields using soil sampler. Soil cores from plots were mixed thoroughly.

The material was mineralized with a CEM Mars 5 Xpress (CEM, Matthews, NC, USA) microwave mineralization system (55 mL vessels) using 8 mL HNO_3_ (65%) and 2 mL H_2_O_2_, according to the program comprising three stages: First stage—power 800 W, time 10 min, temperature 120 °C; second stage-power 1600 W, time 10 min, temperature 160 °C; third stage-power 1600 W, time 10 min, temperature 200 °C [21]. Materials after digestion were filtered through 45 mm filters (Qualitative Filter Papers Whatman, Grade 595: 4–7 µm; GE Healthcare, Buckinghamshire, UK), and filtrate completed with deionized water from Milli-Q Academic System (non-TOC (Total Organic Carbon); Millipores. A.S., Molsheim, France) to a final volume of 50 mL. Concentration of particular trace elements was analyzed by the flame atomic absorption spectrometry (Cd, Cu, Mn, Cr, Co, Si, Ni, and Zn), atomic emission spectrometry (Mg, Ca, Na, K, B) using an AA Duo—AA280FS/AA280Z spectrometer (Agilent Technologies, Mulgrave, Victoria, Australia), equipped with a Varian hollow-cathode lamp (HCL; Varian, Mulgrave, Victoria, Australia). Calibration curves were prepared in four replicates per each trace element concentration. Detection limit for the analyzed metals was, mg kg^−1^: Ca 0.015, Na 0.10, K 0.09, Mg 0.003, B 0.06, Cu 0.18, Zn 0.06, Cr 0.005, Mn 0.005, Co 0.011, Si 0.12, Ni 0.005, Cd 0.01.

### 2.3. Seed Quality

For the evaluation of germination ability, 3 × 50 seeds from each experimental plot (180 samples) were sown in plastic boxes between two layers of moistened (to 60% WR) filter paper. After sowing, the samples were prechilled at 7 °C for 3 days and placed in Sanyo growth chamber (Sanyo Electric Co., Ltd., Osaka, Japan) at constant temperature 20 °C. After four days, first count (germination energy) was made. The normal seedlings were counted and share in percent was evaluated. According to present International Seed Testing Association Rules [22] after eight days, the final count (germination capacity) and evaluation of normal seedlings, abnormal seedlings (AS), dead seeds (DS), and fresh ungerminated (FUS) seeds were made.

### 2.4. Fusarium DNA Quantification with Real-Time PCR

#### 2.4.1. Isolation of Total DNA from Grain

DNA was extracted according to Doyle and Doyle [23] protocol with modifications of Department of Phytopathology and Molecular Mycology UTP.

Ten grams of grain was homogenized to fine powder and 100 mg of such prepared sample was taken for DNA isolation. Samples were transferred into 2.0 mL tubes and poured with 600 μL of the extraction buffer containing CTAB 5.0%, EDTA 0.5 M, NaCl 5.0 M, Tris-HCl (pH 8.0) 1.0 M, β-mercaptoethanol, PVP (2.0%), and water. DNA was purified by addition of phenol:chloroform: isoamyl alcohol mixture (25:24:1) followed by centrifugation and taking the upper phase (supernatant) to the new tube, where equal volume of chloroform:isoamyl alcohol mixture (24:1) was added, mixed by inverting and centrifuged. Supernatant was taken and DNA was precipitated with cold ethanol. DNA pellet was washed with 70% cold ethanol, left to dry for 25–30 min and poured with TE buffer or sterile water to dissolve. Samples were stored at −20 °C for further analyses.

#### 2.4.2. Preparation of Standard Curve

Material for preparation of standard curve was a series of 10-fold dilutions of DNA isolated from pure culture of researched six *Fusarium* species (*F. avenaceum, F. culmorum*, *F. graminearum, F. langsethiae, F. poae* and *F. sporotrichioides*). Pure fungal cultures were grown on PDA medium (Difco, Becton, Dickinson and Company Sparks, MD, USA) on Petri dish and DNA was isolated from scraped and lyophilized mycelium using the same protocol as for grain.

#### 2.4.3. Preparation of DNA Samples for Real-Time PCR

Concentrations of DNA obtained from kernels were measured with Quantus fluorometer (Promega, Madison, WI, USA). All samples were diluted in sterile deionized water to 10 ng∙µL^−1^. Final concentration of *Fusarium* DNA in a sample was expressed in picograms per 100 ng of total DNA.

#### 2.4.4. Real-Time PCR Reaction Conditions

Amplification was performed with LightCycler 480II (Roche, Basel, Switzerland) using LightCycler 480 SYBR Green I Master (Roche, Basel, Switzerland) in a volume of 10 mL per sample (5.5 μL premix + 4.5 μL DNA) in 45 cycles according to thermal profiles specific to each *Fusarium* species. The primers used for each researched *Fusarium* species were shown in Table 2.

### 2.5. Analysis of Trichothecenes

Grain samples (60) were analyzed for the presence of trichothecenes according to Perkowski et al. [28]. Subsamples (10 g) were extracted with acetonitrile:water (82:18) and purified on a charcoal column (Celite 545/charcoal Draco G/60/activated alumina neutral 4:3:4 (*w*/*w*/*w*).

Type A trichothecenes (HT-2 toxin (HT-2), T-2 toxin (T-2), T-2 tetraol, T-2 triol, diacetoxyscirpenol (DAS), scirpentriol (STO)) were analyzed as TFAA (trifluoroacetic anhydride) derivatives. To the dried sample, 100 μL of trifluoroacetic acid anhydride were added. After 20 min, the reacting substance was evaporated to dryness under nitrogen. The residue was dissolved in 500 μL of isooctane and 1 μL was injected onto a gas chromatograph-mass spectrometer (GC/MS, Hewlett Packard GC 6890, Waldbronn, Germany). Type B trichothecenes (DON, NIV, 3-acetyldeoxynivalenol (3-AcDON), 15-acetyldeoxynivalenol (15-AcDON), fusarenon X (FUS-X)) were analyzed as TMS (trimethylsilylsilyl ethers) derivatives. To the dried extract, the amount of 100 μL of TMSI/TMCS (trimethylsilyl imidazole/trimethylchlorosilane; *v*/*v* 100/1) mixture was added. After 10 min 500 μL of isooctane were added and the reaction was quenched with 1 mL of water. The isooctane layer was used for the analysis and 1 μL of the sample was injected on a GC/MS system.

The analyses were run on a gas chromatograph (Hewlett Packard GC 6890, Waldbronn, Germany) hyphenated to a mass spectrometer (Hewlett Packard 5972 A, Waldbronn, Germany), using an HP-5MS, 0.25 mm × 30 m capillary column. The injection port temperature was 280 °C, the transfer line temperature was 280 °C and the analyses were performed with programmed temperature, separately for type A and type B trichothecenes. The type A trichothecenes were analyzed using the following programmed temperatures: Initial 80 °C held for 1 min, from 80 °C to 280 °C at 10 °C min^−1^, the final temperature being maintained for 4 min. For the type B trichothecenes initial temperature of 80 °C was held for 1 min, from 80 °C to 200 °C at 15 °C min^−1^ held for 6 min and from 200 °C to 280 °C at 10 °C min^−1^, with the final temperature being maintained for 3 min. The helium flow rate was held constant at 0.7 mL min^−1^.

Quantitative analysis was performed in the single ion monitored mode (SIM) using the following ions for the detection of STO: 456 and 555; T-2 tetraol 455 and 568; T-2 triol 455 and 569 and 374; HT-2 455 and 327; T-2 327 and 401. DON: 103 and 512; 3-AcDON: 117 and 482; 15-AcDON: 193 and 482; NIV: 191 and 600. Qualitative analysis was performed in the SCAN mode (100–700 amu). Recovery rates for the analyzed toxins were as follows: STO 82 ± 5.3%; T-2 triol 79 ± 5.1%; T-2 86 ± 3.8%; T-2 tetraol 88 ± 4.0%; HT-2 91 ± 3.3%; DON 84 ± 3.8%; 3AcDON 78 ± 4.8%; 15 AcDON 74 ± 2.2%; and NIV 81 ± 3.8%. The limit of detection was 0.01 µg kg^−1^.

### 2.6. Chemical Analysis of Ergosterol

Ergosterol (ERG) in 60 grain samples was determined by HPLC as described by Young [29] with modifications [30,31]. A detailed evaluation of the method was given by Perkowski et al. [31]. Samples containing 100 mg of ground grains were placed into 17-mL culture tubes, suspended in 2 mL of methanol, treated with 0.5 mL of 2 M aqueous sodium hydroxide and tightly sealed. The culture tubes were then placed within 250-mL plastic bottles, tightly sealed and placed inside a microwave oven (Model AVM 401/1WH, Whirlpool, Sweden) operating at 2450 MHz and 900 W maximum output. Samples were irradiated (370 W) for 20 s and after about 5 min for an additional 20 s. After 15 min the contents of culture tubes were neutralized with 1 M aqueous hydrochloric acid, 2 mL MeOH were added and extraction with pentane (3 × 4 mL) was carried out within the culture tubes. The combined pentane extracts were evaporated to dryness in a nitrogen stream. Before analysis samples were dissolved in 4 mL of MeOH, filtered through 13-mm syringe filters with a 0.5 mm pore diameter (Fluoropore Membrane Filters, Millipore, Ireland) and evaporated to dryness in a N_2_ stream. The sample extract was dissolved in 1ml of MeOH and 50 µL were analyzed by HPLC. Separation was performed on a 150 × 3.9 mm Nova Pak C-18, 4 mm column and eluted with methanol/acetonitrile (90:10) at a flow rate of 0.6 mL min^−1^. Ergosterol was detected with a Waters 486 Tunable Absorbance Detector (Milford, MA, USA) set at 282 nm. The presence of ERG was confirmed by a comparison of retention times and by co-injection of every tenth sample with an ergosterol standard.

### 2.7. Statistics

The statistical analysis was performed using Microsoft^®^ Excel 2016/XLSTAT© Ecology (Version 18.0738413, Addinsoft, Inc., Brooklyn, NY, USA). Differences between variable means for the two experimental variants were compared using parametric Student’t t-test (XLSTAT procedure: Two-sample t and z tests). Variables distribution in samples from the two experimental variants were compared using the Kruskal–Wallis one-way analysis of variance (XLSTAT procedure: Comparison of k samples—Kruskal–Wallis, Friedman). The Kruskal–Wallis test was selected because some of the variables did not follow normal distribution.

The relationships between FHBi, seed quality and concentration of ergosterol, mycotoxins and *Fusarium* DNA were investigated by Pearson correlation tests (XLSTAT procedure: *Correlation tests*). Prior to analysis, data that did not follow normal distribution was log10 transformed to normalize residual distributions. Multivariate data analysis method was applied to the data on FHB (FHBi, DS, mycotoxin concentrations, *Fusarium* DNA concentrations) resistance. Principal component analysis (XLSTAT procedure: Principal Component Analysis PCA) was used to show how wheat cultivars within two experimental variants (60 observations) are distributed with respect to the main variation described in the first two components and how variables (FHBi, DS, ERG, sum of type A trichothecenes, sum of type B trichothecenes, *Fusarium* DNA) influence the construction of the two components. PCA results also revealed associations among variables measured by the angle between variable vectors.

Differences between two variants for all variables were analyzed using multidimensional tests (XLSTAT procedure: Multidimensional tests (Mahalanobis)) and multivariate analysis of variance (XLSTAT procedure: *MANOVA*).

Cultivars in the organic field were grouped according to their resistance to infection of heads with *Fusarium* fungi measured by FHB index, dead seeds proportion, ERG, sum of type A trichothecenes, sum of type B trichothecenes, *Fusarium* DNA. K-means clustering procedure of XLSTAT was applied. Results were visualized using Discriminant analysis procedure of XLSTAT. Classes obtained from K-means analysis were applied as a qualitative depended variable in DA analysis.

## 3. Results

### 3.1. Concentration of Mineral Elements in Soil

In order to determine soil conditions in both experimental fields analysis of a number of elements that occur in these environments was made (Appendix A). For the most of 13 analyzed compounds significant differences between organic and conventional fields were found. Only for Co, concentration difference was not significant. In soil of conventional field, concentration of K, Mg, Cd, Cr, Cu, Ni, and Zn was higher than in soil of organic field. The highest differences were found for Zn (7-fold) and Cd (3-fold). On the other hand, in soil of organic field, concentration of Ca, Na, Si, B, and Mn was higher than in soil of conventional field.

### 3.2. Phenotypic Data and Fungal Diseases

Wheat cultivars differed in heading and flowering time. In the conventional field, heading time was 29.7 days from 1 May, at a range 24.0 (‘Smuga’)—35.0 days (‘Sukces’) (Appendix A). Flowering time was on average 31.6 days from 1 May, at a range 26.0 (‘Smuga’)—35.0 days (‘Sukces’, ‘Boomer’). In the organic field, heading time was 28.0 days from 1 May, at a range 24.0 (‘Smuga’)—33.0 days (‘Sukces’). Flowering time was on average 29.8 days from 1 May, at a range 26.0 (‘Smuga’, ‘Ludwig’)—35.0 days (‘Sukces’). Heading and flowering time were significantly earlier in organic field than in conventional one (Table 3).

On average, plant height of wheat cultivars did not differ between organic and conventional fields (Table 3). In organic field plant height ranged between 73.7 cm (‘Muszelka’) and 114.3 cm (‘Akteur’) (Appendix A). In conventional field, this parameter ranged between 76.3 cm (‘Alcazar’) and 118.7 cm (‘Ludwig’).

Fusarium head blight severity was low and average values for conventional and organic fields did not differ significantly (Table 3, Figure 1). In conventional field, FHB index range was from 0 to 4.4%. Cultivars ‘Nateja’ and ‘Legenda’ showed no symptoms of FHB and cultivars ‘Kampana’, ‘Muszelka’ and ‘Slade’ were the most infected (FHBi = 4.4%, 3.5%, and 2.1%, respectively) (Appendix A). In organic field, FHB index range was from 0 to 3.2%. Cultivars ‘Nateja’ and ‘Mewa’ showed no symptoms of FHB and cultivars ‘Slade’, ‘Kampana’, ‘Turkis’ and ‘Belenus’ were the most infected (FHBi = 3.2%, 2.3%, 1.8%, and 1.8%, respectively). FHB indexes for conventional and organic fields correlated significantly (*r* = 0.776 at *p* < 0.001).

Heading and flowering time were not correlated with FHB severity. In both fields plant height significantly negatively correlated with FHB indexes (*r* = −0.519 for organic field and *r* = −0.589 for conventional field; at *p* < 0.001).

Symptoms of leaf diseases in both experimental plots were observed starting from half of May, when yellow rust was detected. Seventeen cultivars were fully resistant to yellow rust and showed no symptoms of disease (Appendix A). On average yellow rust severity was slightly higher in conventional field; however, difference with organic field was not significant. Leaf rust severity was low. On average, it was 0.7% in organic field and 1.0% in conventional one. (Appendix A). Symptoms of Septoria tritici blotch were observed on most (28) cultivars in conventional field and on 17 cultivars in organic field. Symptoms of Stagonospora nodorum blotch were observed on 12 cultivars in conventional field and on 23 cultivars in organic field. Tan spot was observed only in organic field with average severity 2.0%. This disease affected fourteen cultivars.

Grain yield per plot was similar for both field and do not differ statistically significantly (Table 3). In organic field grain yield ranged between 3.1 kg (‘Nateja’) and 6.7 kg (‘Jenga’) (Appendix A). In conventional field, this parameter ranged between 2.9 kg (‘Nateja’) and 6.6 kg (‘Boomer’). In both field grain yield was significantly negatively correlated with yellow rust severity (*r* = −0.517 for organic field and *r* = 0.647 for conventional field; at *p* < 0.001) and not correlated with FHB indexes.

### 3.3. Characteristic of Seed Germination

Sowing quality of seeds from conventional field was significantly higher than those from organic one were (Table 4). The mean value for the germination energy for the conventional seeds was much higher (87%) than for the organic seeds (63.2%). Similar was found for the germination capacity. In organic seed material lower percent’s share of normal seedlings, but higher number of abnormal seedlings, dead seeds and fresh ungerminated seeds was observed.

Values of the germination energy ranged from 54.0% (‘Mewa’) to 98.0% (‘Nateja’) in conventional samples and from 11.5% (‘Belenus’) to 93.0% (‘Mewa’) in organic samples (Appendix A). The germination capacity ranged from 83.5% (‘Slade’) to 98.5% (‘Nateja’) in conventional material and from 75.0% (‘Belenus’) to 96.0% (‘Batuta’) in organic material. The percent shares of abnormal seedlings as well as dead seeds were significantly higher in organic samples (Table 4). These variables ranged from 0.5% (‘Markiza’) to 8.5% (‘Alcazar’, ‘Garantus’) and from 0.5% (‘Batuta’) to 15.0% (‘Belenus’) in organic field and 0 (‘Figura’, ‘Belenus’) to 7.5% (‘Mewa’, ‘Ostroga’, ‘Slade’) and from 0 (‘Batuta’, ‘Nateja’) to 7.5% (‘Jenga’) in conventional field. Additionally, percentage of fresh, ungerminated seeds was twice higher in organic material than in conventional. However, difference was not significant. It was the highest in organic seed material of cultivars ‘Belenus’ (7.5%), ‘Akteur’ (6.0%), and ‘Ostroga’ (5.0%).

### 3.4. Concentration of Ergosterol and Trichothecenes

Concentration of ergosterol in grain was significantly higher in samples from organic field than from conventional one (Table 5). Level of ERG varied from 0.26 (‘Ostka Strzelecka’) to 1.85 mg kg^−1^ (‘Mulan’) in conventional field and from 0.26 (‘Boomer’) to 3.46 mg kg^−1^ (‘Akteur’) in organic field (Appendix A).

Amount of type B trichothecenes was low and varied from 8.9 to 460.2 μg kg^−1^ in conventional field and from 10.1 to 384.5 μg kg^−1^ in organic field (Table 5). On average, more trichothecenes were present in grain from conventional field; however, difference was statistically insignificant. Regarding specific toxins, only concentration of 3-AcDON was significantly higher in conventional samples. Concentration of NIV was higher in samples from organic field; however, difference was not significant. Distributions for FUS-X and 3-AcDON in organic and conventional samples were significantly different. In conventional samples, these toxins were detected in higher amounts in single samples whereas they were more evenly distributed in organic samples.

The highest concentrations of type B trichothecenes were found in grain of cultivars ‘Anthus’, ‘Ostroga’, and ‘Garantus’ in conventional field (460.2 μg kg^−1^, 321.1 μg kg^−1^, 308.9 μg kg^−1^, respectively) and in grain of ‘Alcazar’, ‘Kampana’, ‘Muszelka’ and ‘Anthus’ (384.5 μg kg^−1^, 278. μg kg^−1^, 257.6 μg kg^−1^, 244.6 μg kg^−1^, respectively) in organic field (Appendix A).

Amount of type A trichothecenes was very low and similar in conventional and organic samples (5.5 and 5.1 μg kg^−1^, respectively) (Appendix A). It varied from 0.7 (‘Boomer’) to 30.5 μg kg^−1^ (‘Garantus’) in conventional samples and from 1.0 (‘Figura’) to 14.5 μg kg^−1^ (‘Zyta’) in organic samples. Average concentrations of type A trichothecenes were similar in both groups, and they not differ significantly. Only average concentration of DAS was significantly higher in conventional samples.

### 3.5. Fusarium Species

Presence of biomass of six *Fusarium* species was detected in wheat grain. *Fusarium langsethiae* was detected only in six samples in trace amounts. On average, the highest amount of DNA was found as follows for *F. poae*, *F. graminearum*, *F. sporotrichioides*, *F. culmorum,* and *F. avenaceum* (Table 6). It was true in organic field. In conventional field concentration of *F. culmorum* DNA was higher than *F. graminearum* and *F. sporotrichioides* DNA.

Total *Fusarium* DNA concentration in organic samples was more than twice higher than in conventional samples. It ranged from 15.4 (‘Batuta’) to 405.2 pg 100 ng^−1^ (‘Figura’) in conventional samples and from 15.3 (‘Batuta’) to 1205.8 pg 100 ng^−1^ (‘Slade’) in organic samples. The difference was statistically significant. Similarly, significantly higher (about three times) were concentrations of *F. poae, F. graminearum, F. avenaceum,* and *F. sporotrichioides* in organic samples. Amount of *F. culmorum* was about twice higher in organic samples; however, difference was not statistically significant. The most *Fusarium* colonized was grain of ‘Figura’, ‘Kampana’, and ‘Alcazar’ cultivars in conventional field and ‘Muszelka’, ‘Kampana’, ‘Turkis’, ‘Ostroga’, ‘Meteor’, ‘Bogatka’, ‘Alcazar’, and ‘Slade’ in organic field (Appendix A).

### 3.6. Correlation Between Experimental Components

Result’s correlations of the evaluation of conventional wheat material showed that the proportion of the dead seed was poorly correlated with the germination energy, but highly negatively correlated with germination capacity (Table 7). In the case of organic seed, the dependence was the same, except that the negative values of the correlation coefficients were higher (Table 8). The proportion of abnormal seedlings was significantly negatively correlated with the energy and germination capacity, as well as the number of dead seeds in conventional material. In contrast to the results of organic material, where the relevant dependencies for these traits were not found. However, organic material has demonstrated a highly significant negative relationship between the proportion of fresh ungerminated seeds and the energy and germination capacity and a positive correlation between fresh ungerminated and dead seed.

In conventional and ecological material, the same negative relationship between FHBi and germination was found, and the positive relationship between FHBi and the share of dead seeds occurred. These two parameters in conventional samples correlated also significantly with concentration of type B trichothecenes in grain.

We observed significant effect of colonization of kernels with *Fusarium* species on seed quality (Table 7 and Table 8). The proportion of dead seed in conventional material was highly correlated with the quantity of the DNA of *F. graminearum*, while in organic material with the amount of DNA of *F. poae*, *F. sporotrichioides,* and total *Fusarium* DNA.

There was lack of correlation between Fusarium head blight index and amount of ERG and type A and B trichothecenes in grain in both variants (Table 7 and Table 8). However, in conventional samples positive tendency FHBi versus type B trichothecenes was observed and the same was found for FHBi versus type A trichothecenes in organic samples. Fusarium head blight index correlated significantly with concentration of DNA of three *Fusarium* species—*F. avenaceum*, *F. graminearum,* and *F. poae* in both variants. In organic variant FHBi correlated significantly also with *F. sporotrichioides* DNA concentration. No correlation was found with *F. culmorum* DNA. Ergosterol content in grain did not correlated with type A or B trichothecenes as well as with DNA concentration of *Fusarium* species.

In samples from conventional field amount of type B trichothecenes correlated highly significantly with *F. graminearum* DNA but not with *F. culmorum* DNA. Contradictory, in organic samples *F. graminearum* did not correlate with type B trichothecenes and for *F. culmorum* there was found positive tendency however not statistically significant. Regarding specific toxins in conventional samples *F. graminearum* correlated significantly with DON amount (*r* = 0.531) and for *F. culmorum* some positive tendency was observed for FUS-X and 15-AcDON. In organic samples, only correlation of *F. culmorum* with 3-AcDON (*r* = 0.421) was found. There was no significant correlation between amount of type A trichothecenes and potentially producing species *F. sporotrichioides* and *F. poae*.

Amounts of DNA of three *Fusarium* species (*F. avenaceum*, *F. graminearum* and *F. poae*) in grain form both variants correlated statistically significantly (Table 7 and Table 8). *Fusarium sporotrichioides* DNA concentration correlated with *F. avenaceum* and *F. poae DNA*. *Fusarium culmorum* DNA concentration did not correlated with the other species.

### 3.7. Multivariate Principal Component Analysis

Multivariate principal component analysis showed significant difference between two studied populations (wheat cultivars in two environments) in terms of FHB infection (Figure1). However, this difference was caused by only some cultivars, which showed higher *Fusarium* infection (measured with different parameters) in organic or conventional field. Cultivars from organic field had higher FHB index, proportion of dead seeds and *Fusarium* DNA content. In conventional field, the most infected cultivars had higher toxin content in the grain but moderate FHB index, dead seeds proportion and *Fusarium* biomass amount in kernels. The exception was cultivar ‘Kampana’ (C_12) (Figure 2).

There were also carried out other tests—Multidimensional Wilks’ Lambda test and Fisher distances test. They pointed to the significance of the separation between the analyzed growing systems at the significance level of *p* < 0.0001.

There was also compared which source of variation had higher effect on the obtained results (i.e., FHBi, DS, Total TCT B, Total TCT A, and *Fusarium* DNA concentration) using multivariate analysis of variance (MANOVA). Both sources statistically significantly affected the results; however, experimental variant (conventional vs organic field) had much higher significance (*p* < 0.0001). It means that Fusarium head blight infection and its effect on grain quality, toxins concentration and *Fusarium* biomass in kernels depended mainly on wheat growing environment. Resistance of cultivars to FHB was less important (*p* < 0.025).

Cultivars grown in organic field were compared for their overall performance under such conditions with respect to resistance to *Fusarium* infection. Multivariable analysis (K-means, discriminant analysis) made it possible to divide cultivars into three groups depending on their resistance to head infection, number of dead seeds, accumulation of ergosterol and *Fusarium* toxins in the grain as well as contamination of grain with *Fusarium* fungi (Figure 3, Table 8).

The most infected five cultivars were in the second group (Figure 3, Table 9). They could be described by the highest FHB index, high number of dead seeds, high accumulation of *Fusarium* toxins and the highest concentration of *Fusarium* biomass in kernels. Only amount of ERG was medium in grains of the cultivars of the group 2. The other 25 cultivars were in two close groups 1 and 3. They mainly differed in amount of ERG in grain, which was the highest in the group 1 while the lowest in the group 3.

The lowest overall infection showed cultivars from the group 3: ‘Nateja’, ‘Mewa’, and ‘Markiza’. Regarding only accumulation of trichothecenes, it was the lowest in grain of ‘Nateja’ and ‘Ostroga’ from the group 3. It was low also in grain of cultivars ‘Figura’, ‘Dorota’, ‘Mulan’, ‘Batuta’, ‘Tonacja’ from the group 1, and surprisingly in grain of cultivar ‘Belenus’ from group of the most infected cultivars. Among the low-toxin accumulating cultivars, ‘Ostroga’, ‘Batuta’, ‘Mulan’, and ‘Figura’ were in the group of cultivars showing the highest grain yield per plot in organic field (Appendix A). The lowest infected cultivar ‘Nateja’ had low grain yield caused by high yellow rust infection.

## 4. Discussion

Until recently, the issue of organic farming was considered marginal. However, the constantly increasing acreage crops grown in this system and fast increasing percentage of consumers interested in obtaining the organic food encouraged a detailed address of this issue, which, so far, was recognized only partially. Thus, it was decided to provide field experiment for a representative sample of winter wheat cultivars for both systems of crops growing-conventional and organic under the same environmental conditions (location, time, and weather). Of course, not all growing conditions were the same. This was mainly related to soil conditions. This was reflected in the studies presented by the analysis of the mineral elements. In most cases, we found significant differences in the concentrations occurring in the soil. A higher concentration in organic soil was found for Ca, Na, Si, B, and Mn. Lower for K, Mg, Cd, Co, Cr, Cu, Ni, and Zn. This is certainly due to organic cultivation resulting from the lack of use of mineral fertilizers. This leads to the relative impoverishment of the soil. However, we found higher concentration of Ca, Na, B and Si in soil from organic field. In review paper by Romero et al. [32] the authors found that silicon shows the beneficial effects on growth, development, and health of crops. It activates the defense mechanisms of plants and increases tolerance to fungal diseases [33]. Concentrations of all analyzed metals (despite Mn) was lower in organic soil. Differences were significant for Cr, Cu, Ni, and Zn. Depletion of the soil in organic cultivation system has an impact on the amount of soil microorganisms [34,35]. They are responsible for the biochemical processes, and thus, consequently, the resistance mechanism of plants. Studies on the reduction of the content of alkaline metals showed it leads to its acidification, which promotes the growth of micro-organisms for which the acidic (pH 4–5) is beneficial, including microscopic fungi and among them plant pathogenic species [36,37]. With this widespread phenomenon, we have to deal with in our research. These confirms our results indicating a higher concentration of K, Mg, Zn, or Cd detected on fertilized, conventional plots. Much higher concentration of Zn and Cd was especially interesting.

The presented experiments concerned 30 cultivars of winter wheat, which were examined comprehensively for several years under conventional conditions to determine their susceptibility to Fusarium head blight and were divided into 4 groups as shown in Table 1 [20,38]. They constitute a complete cross-section of widely grown wheat cultivars in Poland, which gives the basis for determining them as model cultivars. In conditions of experimental year, Fusarium head blight severity was low and average values for conventional and organic field did not differ significantly. Heading and flowering time were significantly earlier in organic field than in conventional. On average, plant height of wheat cultivars did not differ between two analyzed cropping systems. Effect of plant height of FHB severity was cultivar-depended and similar in both systems. Taller cultivars were less FHB infected.

Sowing quality measured as germination energy and germination capacity of conventional material was significantly higher than for the seeds from organic cultivation system. The percentage’s share of abnormal seedlings as well as share of dead seeds was significantly higher in organic seed material. It confirms results obtained for sowing value of conventional and organic oats [39]. Additionally, percentage’s share of fresh ungerminated seeds was twice higher in organic seed material than in conventional one. However, difference was not statistically significant.

It is evident that there were significant differences in the seed quality obtained from both cultivation systems. For all parameters, the differences were important to the detriment of organic farming. Particularly large was the difference in germination energy, which for material from the ecological field was almost 25% lower than for the conventional. The above differences may also result from higher colonization of kernels by mycobiota obtained from the ecological field. This was indicated by twice-higher ergosterol content (the total fungal quantity meter) and a three-fold higher *Fusarium* biomass content in organic grain. The content of ergosterol in grain depends on the type of grain (hulled, hull-less), cereal species and on the level of contamination of the grain with microscopic fungi, both pathogenic strains and native mycobiota [31,40,41]. Its content is affected also by the method of cultivation resulting from the use of fertilizers and treatments related to the use of plant protection chemicals [30].

Plant protection causes a disturbance of the natural homeostasis of the microbiota of kernel’s surface, resulting in development of more expansive microbes that can dominate the environment. Thus, unfortunately, probiotic microorganisms, which are a natural barrier to pathogens, are completely removed [42,43]. The logical consequence is that, because of conventional agro-technical management, such crop is more susceptible to colonization by mycobiota. On the other hand, this increases competitiveness with pathogenic fungi producing specific fungal metabolites. Consequently, this has to do with the detection of a twice-higher concentration of ergosterol in organic material. Among the detected types of microscopic fungi, pathogenic ones represent a small percentage. This is related to the presence of large amounts of nonpathogenic mycobiota, which is a competition for pathogens [44,45].

Presence of DNA of six *Fusarium* species was detected in wheat grain. *F. langsethiae* was detected in six samples only in trace amounts. The highest amount of DNA was found as follows for *F. poae*, *F. graminearum*, *F. sporotrichioides*, *F. culmorum* and *F. avenaceum*. It was true in organic field. In conventional field, concentration of *F. culmorum* DNA was higher than *F. graminearum* and *F. sporotrichioides* DNA. The composition of *Fusarium* species was similar to that observed in last years in Europe [5,46,47,48].

Total *Fusarium* DNA concentration in organic samples was more than twice higher than in conventional samples, what can be explained by the cultivation system. In the case of organic cultivation, an environmental niche with a stabilized microorganism population is formed, which is enriched with probiotic organisms. Living in a symbiosis of microorganisms contribute to the improvement of soil condition, and thus naturally strengthen the resistance mechanisms of plants through i.a. mycorrhiza. In the conventional case, there are stress related to fertilization or the use of pesticides. Some fungi are eliminated, others often having a strong pathogenic remain. An analysis of the DNA content in the grain also gives a lot of interesting information. In the grain of organic farming, almost three times more DNA was found and stronger links between the contents of single species were identified. This was not true for *F. culmorum*, which could be related to the presence of *F. graminearum* in grain species being a competitor in the biosynthesis of type B trichothecenes.

Another legitimate conclusion here is that the testing of the DNA content of the grain is a much more accurate test method than the determination of the fungal concentration by an ERG analysis [49,50,51]. At the same time, it is emphasized that the amount of ERG gives a full image of the level of contamination with microscopic fungi. This is confirmed by the correlation factors for the ERG. They are, in all cases insignificant what confirms the above argument. Pathogenic fungi in the grain produce various metabolites and among them mycotoxins. This also occurs in the case of fungi of the genus *Fusarium*, which synthesize trichothecene toxins. This was also the case in the analyzed samples.

Based on the above results and conclusions is imposed another one. In grain of organic farming theoretically, the concentration of trichothecenes should be significantly higher. However, it was found that type A and B trichothecenes concentrations in both cases were similar and the differences were not significant. Vanova et al. [52] found higher concentration of DON in grain of wheat grown in three conventional systems. It was significantly higher in two systems where no chemical protection against FHB was applied. Similar tendency was found in barley and oats from organic and traditional farming [53,54,55,56]. In their review, Brodal et al. [56] concluded that contamination with *Fusarium* toxins of organically produced cereal grains was similar and sometimes lower than conventionally produced ones.

The established correlation coefficients for both groups were significant for the conventional system only. This is probably due to the fact, that for this system, more fungal biomass of *F. graminearum* was found in the grain. That resulted in a higher correlation with the sum of type B trichothecenes which *F. graminearum* is an important producer.

The concentration of detected toxins was relatively small. Concentration of DON and T-2/HT-2 toxins was below the European limit and recommendation (Commission Regulation No. 1126/2007 of 28 September 2007; Commission Recommendation No. 2013/165/EU of 27 March 2013). Comparing the two cultivation systems, however, it is evident that in grain the average concentration of type B trichothecenes was lower in the case of organic trials. Differences were not statistically significant, but at the same time, concentration of *Fusarium* DNA was almost 3 times higher in organic grain. Although it can be also found similar data in other papers [52,57,58,59,60] it is a positive result. This proves once again that in the organic system determines the community of co-existing microorganisms is established. The most pathogenic and toxigenic are not predominant and environmental stress is not as harmful as the stress associated with significant doses of artificial fertilizers and pesticides as well as simplified rotations [61,62].

When considering the concentration of specific toxins, only concentration of 3-AcDON was significantly higher in conventional samples. Concentration of NIV was higher in samples from organic field; however, difference was not significant. Distributions for FUS-X and 3-AcDON in organic and conventional samples were significantly different. In conventional samples, these toxins were detected in higher amounts in single samples whereas they were more evenly distributed in organic samples. Amount of type A trichothecenes was very low and similar in conventional and organic samples, and they not differ significantly. Only average concentration of DAS was significantly higher in conventional samples.

Comparison of the sum of trichothecene toxins of groups A and B indicate environmental effects. Important correlations were obtained for the conventional system, by the fact that a strong pathogenic species *F. graminearum* stood out, being the most important producer of such toxins as DON, its derivatives and to a lesser extent (depending on the chemotype) NIV. For organic farming, the established coexistence of species was confirmed and no dominance of *F. graminearum* was found. When analyzing occurring mycobiota using ERG as a measure, no significant correlation was found for both environments with the other characteristics. In the case of *Fusarium* DNA testing, such correlations were found with the stronger link found for organic farming both between species (except *F. culmorum*) and other studied traits (mainly FHBi). The data presented is a significant contribution to understanding the philosophy of cultivation system and its effects. A similar method of reasoning and application may be found in paper of Lazzaro et al. [63].

By summarizing this aspect of the research, it is possible to identify clearly the relationship between the analyzed factors in the case of organic cultivation as stronger (Table 7). The above statements was confirmed by a comprehensive statistical analysis. It included a number of tests comparing analyzed populations based on factors such as FHBi, DS, ERG, Total TCT A and B, and *Fusarium* DNA concentration. The designated *p*-value for the multidimensional Wilk’s test had the value < 0.0001. *p*-value was similar for Fisher distances. It gives clear grounds for supporting the above conclusions indicating the different mechanism of reaction of plants on environmental stress of both cultivation systems. The MANOVA test was conducted to further validate these conclusions. It clearly showed that its effects depended on the type of cropping system in a very important way (*p* < 0.0001), and to a much lesser extent on the cultivars used in the experiment (*p* < 0.025). Similar observations can be found in the work of e.g., Newton et al. [64].

The issue of wheat cultivars applied in cultivation is often raised [3,4,65,66]. The most important question is whether the same cultivar can be used in both systems. During the study we wanted also to deepen this issue using 30 different cultivars with varying resistance to FHB. The possibility of successfully applying the same cultivar in both systems is becoming increasingly important, also for breeding reasons. Biplot of the principal component analysis shown in Figure 2 indicates the effects of cultivar on the results of the experiment.

It can be concluded that the results indicate a diversified behavior of cultivars, which was characterized by varying distances between cropping effects in two systems. Determined by multidimensional scaling (MDS) method the average distances between pairs in the conventional and organic systems were for resistant cultivars (R) 0.576; medium resistant (MR) 2.335; medium susceptible (MS) 2.819 and susceptible (S) 3.547. This result is unambiguous and indicates that it is possible to use the cultivars used in conventional crops for organic farming [64,67].

The final stage of the study was comparison of the overall performance of cultivars grown under organic field conditions with respect to resistance to *Fusarium* infection. Using multivariable analysis (K-means, discriminant analysis), it was possible to divide cultivars into 3 groups depending on traits tested as indicated in Table 8 and Figure 3. The division on the three groups finds its justification both in the values shown in the table and separation because of their FHB susceptibility. For five cultivars (‘Alcazar’ (S), ‘Muszelka’ (S), ‘Kampana’ (S), ‘Belenus’ (MS), ‘Slade’ (MS)), significantly higher values (excluding ERG) have been obtained for all experimental traits. Discriminant analysis confirmed the condition of these cultivars, which already in other experiments showed low resistance after artificial inoculation of heads when they had high head infection and DON accumulation [20,38].

All the results presented indicate the usefulness of the above studies for the recommendation of individual cultivars to a particular growing method. Such studies requires evidently multiyear or multi location experiments to be fully reliable. Results show differences in effects of the conventional and organic system. The interesting preliminary results obtained, in the meaning of authors, will contribute to a better understanding of the processes of growth and development and effect of cereal farming in certain environmental conditions. They also allow for an objective look at organic farming and perhaps contribute to its rapid growth, as the idea of sustainable cultivation and avoidance of plant stress should gain new supporters.

## Figures and Tables

**Figure 1 microorganisms-07-00439-f001:**
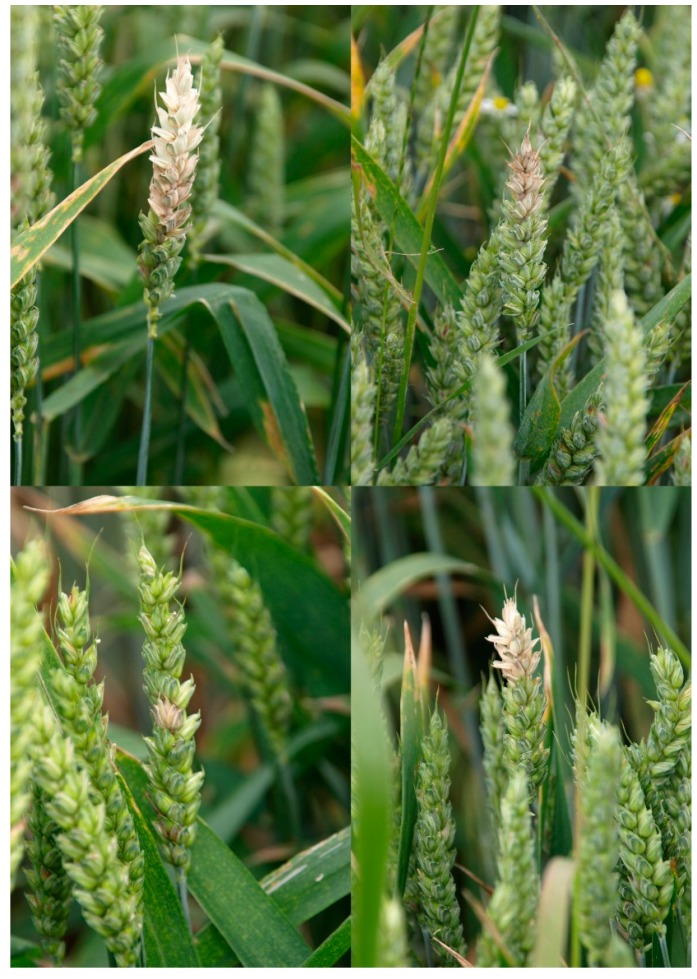
FHB symptoms on heads of wheat grown in organic field. Clockwise from top left: ‘Figura’, ‘Muszelka’, ‘Kampna’, ‘Slade’. Wheat at early milk growth stage (BBCH 73).

**Figure 2 microorganisms-07-00439-f002:**
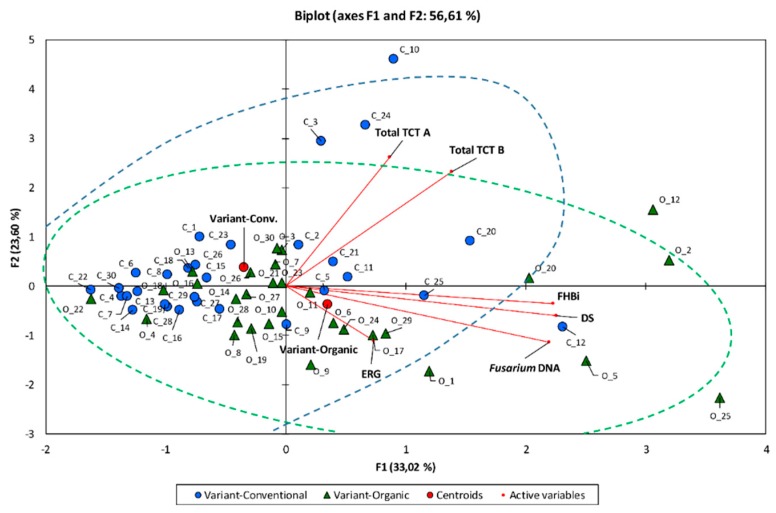
Biplot of the principal component analysis for 30 winter wheat cultivars grown in conventional (C) and organic field (O). Two first components explained 55.90% of variability of Fusarium head blight index (FHBi), dead seeds proportion (DS), ergosterol (ERG) and type A (Total TCT A) and type B (Total TCT B) trichothecenes content, and concentration of DNA (*Fusarium* DNA) of five *Fusarium* species in grain. Samples from conventional field marked with circles and from organic field with triangles.

**Figure 3 microorganisms-07-00439-f003:**
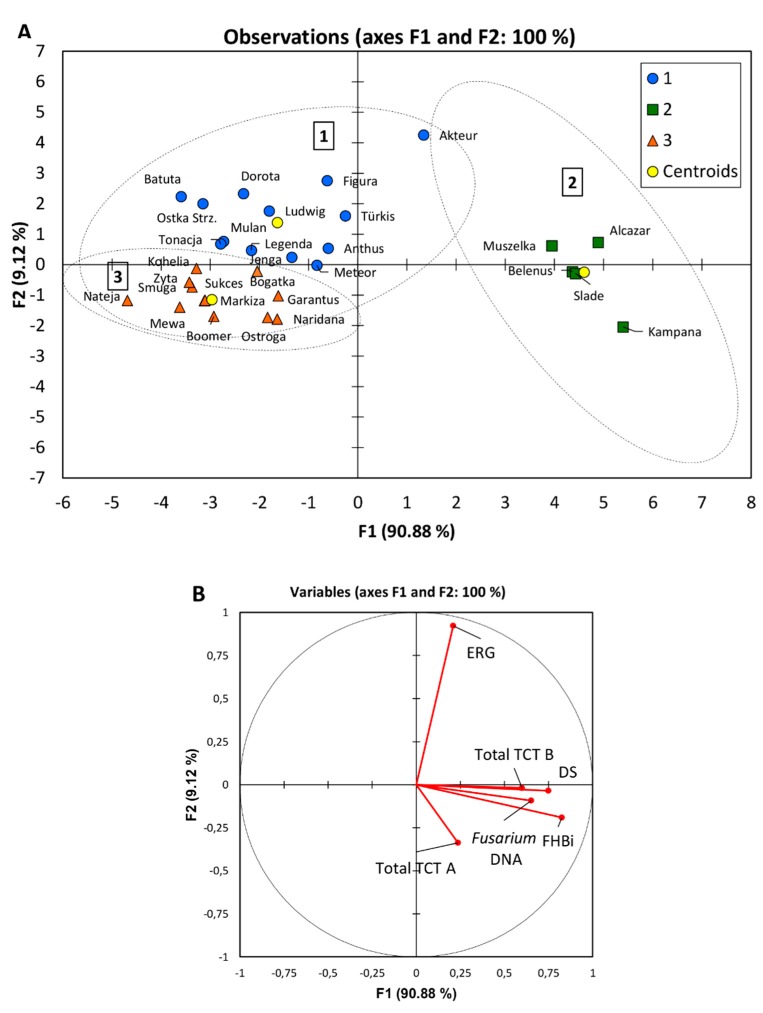
Discriminant analysis of 30 cultivars grown in organic field for Fusarium head blight index (FHBi), dead seeds proportion (DS), ergosterol (ERG), type A (Total TCT A) and type B (Total TCT B) trichothecenes content, and concentration of DNA of five *Fusarium* species in grain (*Fusarium* DNA): (**A**) Observations on the factor axes with marked groups 1–3; (**B**) Correlation circle.

**Table 1 microorganisms-07-00439-t001:** List of winter wheat cultivars used in this study.

No.	Cultivar		No.	Cultivar		No.	Cultivar	
1	Akteur	MS *	11	Jenga	MS	21	Naridana	MS
2	Alcazar	S	12	Kampana	S	22	Nateja	R
3	Anthus	MS	13	Kohelia	MR	23	Ostka Strzelecka	MS
4	Batuta	MS	14	Legenda	MR	24	Ostroga	MR
5	Belenus	MS	15	Ludwig	MS	25	Slade	MS
6	Bogatka	MR	16	Markiza	MS	26	Smuga	S
7	Boomer	MR	17	Meteor	MS	27	Sukces	MR
8	Dorota	MR	18	Mewa	MS	28	Tonacja	MR
9	Figura	MS	19	Mulan	MS	29	Türkis	MS
10	Garantus	MS	20	Muszelka	S	30	Zyta	MR

* Group of resistance to Fusarium head blight [20]; S = susceptible, MS = medium susceptible, MR = medium resistant, R = resistant.

**Table 2 microorganisms-07-00439-t002:** Sequences and names of *Fusarium* species specific primers

*Fusarium* Species	Primer Name	Sequence (5′–3′)	Source
*F. avenaceum*	JIAf	GCTAATTCTTAACTTACTAGGGGCC	[24]
	JIAr	CTGTAATAGGTTATTTACATGGGCG	
*F. culmorum*	Fc01F	ATGGTGAACTCGTCGTGGC	[25]
	Fc01R	CCCTTCTTACGCCAATCTCG	
*F. graminearum*	Fg16F	CTCCGGATATGTTGCGTCAA	[25]
	Fg16R	GGTAGGTATCCGACATGGCAA	
*F. langsethiae*	FlangF3	CAAAGTTCAGGGCGAAAACT	[26]
	LanspoR1	TACAAGAAGACGTGGCGATAT	
*F. poae*:	Fp82F	CAAGCAAACAGGCTCTTCACC	[27]
	Fp82R	TGTTCCACCTCAGTGACAGGTT	
*F. sporotrichioides*	FsporF1	CGCACAACGCAAACTCATC	[26]
	LanspoR1	TACAAGAAGACGTGGCGATAT	

**Table 3 microorganisms-07-00439-t003:** Phenotypic characters, grain yield and Fusarium head blight (FHB) infection of 30 wheat cultivars grown in conventional and organic field.

Variant	Heading (Days from 1st May)	Flowering (Days from 1st May)	Plant Height (cm)	Grain Yield Per Plot (kg)	FHBi (%)
Conventional					
Mean	29.7 b	31.6 b	97.8 a	5.0 a	0.74 a
Std. deviation	2.53	2.39	12.22	0.89	1.00
Organic					
Mean	28.0 a	29.8 a	99.0 a	5.1 a	0.66 a
Std. deviation	2.39	2.55	10.39	0.76	0.77

Values within the same column followed by the different letters are significantly different at the level of probability < 0.01.

**Table 4 microorganisms-07-00439-t004:** Germination characteristic of 30 wheat cultivars grown in conventional and organic field.

Variant	Germination Energy (%)	Germination Capacity (%)	Abnormal Seedlings (%)	Dead Seeds (%)	Fresh, Ungerminated Seeds (%)
Conventional					
Mean	87.0 b ***	93.4 b ***	3.6 a **	2.5 a **	0.6 a
Std. deviation	10.86	3.81	2.26	1.79	0.63
Organic					
Mean	63.2 a ***	89.3 a ***	5.3 b **	4.2 b **	1.2 a
Std. deviation	24.00	4.81	2.10	3.01	1.91

Values within the same column followed by the different letters are significantly different at the level of probability *** *p* < 0.001 or ** *p* < 0.01.

**Table 5 microorganisms-07-00439-t005:** Concentrations of ergosterol (mg kg^−1^) and type B trichothecenes (μg kg^−1^) in grain of 30 wheat cultivars grown in conventional and organic fields.

Variant	ERG	DON	FUS-X	3-AcDON	15-AcDON	NIV	Total TCT B
Conventional							
Mean	0.74 a **	84.8 a	0.9 a	7.3 b ***	1.5 a	5.6 a	100.0 a
Range	0.26–1.85	5.8–444.4	0–11.6	2.3–30.3	0–14.3	0–19.0	8.9–460.2
Std. deviation	0.39	97.3	2.4	5.9	2.6	5.0	101.4
Organic							
Mean	1.42 b **	63.7 a	0.9 a	3.1 a ***	1.1 a	7.4 a	76.2 a
Range	0.26–3.46	2.2–348.4	0–2.9	0–6.2	0–3.3	0–29.5	10.1–384.5
Std. deviation	0.87	86.2	1.0	1.5	1.2	7.3	93.6

Values within the same column followed by the different letters are significantly different at the level of probability *** *p* < 0.001 or ** *p* < 0.01. ERG—ergosterol; DON—deoxynivalenol; FUS-X—fusarenon X; 3-AcDON—3-acetyldeoxynivalenol; 15-AcDON—15-acetyldeoxynivalenol; NIV—nivalenol; Total TCT B— total type B trichothecenes concentrations.

**Table 6 microorganisms-07-00439-t006:** Concentration of DNA (pg 100 ng^−1^ of wheat DNA) of five *Fusarium* species in grain of 30 wheat cultivars grown in conventional and organic fields.

Variant	*F. a.* DNA	*F. c.* DNA	*F. g.* DNA	*F. p.* DNA	*F. sp.* DNA	*Fusarium* DNA
Conventional						
Mean	10.8 a ***	23.3 a	22.7 a ***	34.7 a ***	15,1 a *	106.6 a ***
Range	0–106.7	0–346.2	0.7–79.5	9.4–132.4	0–113.3	15.4–405.2
Std. deviation	19.61	63.14	21.58	24.82	29.10	90.91
Organic						
Mean	30.2 b ***	41.1 a	67.0 b ***	98.2 b ***	50.5 b *	285.7 b ***
Range	0.2–184.6	0–415.5	0.5–280.0	14.1–222.0	0–350.0	15.3–1205.8
Std. deviation	44.88	85.73	62.92	56.44	94.3	244.83

Values within the same column followed by the different letters are significantly different at the level of probability *** *p* < 0.001 or * *p* < 0.05. *F. a.—F. avenaceum*, *F. c.—F. culmorum*, *F. g.—F. graminearum*, *F. p.—F. poae*, *F. sp.—F. sporotrichioides*, *Fusarium* DNA—total DNA of five species.

**Table 7 microorganisms-07-00439-t007:** Coefficients of correlations between seed quality parameters (germination energy—GE, germination capacity—GC, dead seeds, abnormal seedling—AS, fresh ungerminated seeds—FUS), *Fusarium* head blight index (FHBi), ergosterol (ERG) and type A and B trichothecenes concentrations (Total TCT A, Total TCT B) as well as amount of DNA of five *Fusarium* species in grain samples from conventional field.

Variables	GE	GC	Dead Seeds	AS	FUS	FHBi	ERG	Total TCT B	Total TCT A	*F. a.* DNA	*F. c.* DNA	*F. g.* DNA	*F. p.* DNA	*F. sp.* DNA
Final count	0.584 ***													
Dead seeds	−0.419 *	−0.803 ***												
AS	−0.553 **	−0.777 ***	0.436 *											
FUS	−0.112	−0.320	0.048	0.012										
FHBi	−0.125	−0.476 **	0.545 **	0.219	0.058									
ERG	−0.022	0.222	−0.111	−0.190	−0.162	−0.064								
Total TCT B	−0.019	−0.431 *	0.369 *	0.312	0.160	0.290	−0.163							
Total TCT A	−0.216	−0.252	0.185	0.373 *	0.044	0.096	0.011	0.573 ***						
*F. a* DNA	0.258	−0.221	0.412 *	0.095	−0.170	0.430 *	0.016	−						
*F. c* DNA	−0.052	−0.215	0.168	0.002	0.289	0.011	−0.325	0.174	−	−0.013				
*F. g* DNA	−0.204	−0.502 **	0.661 ***	0.288	0.010	0.586 ***	−0.104	0.501 **	−	0.583 ***	0.347			
*F. p* DNA	0.259	−0.169	0.393 *	−0.020	0.010	0.388 *	−0.285	0.202	0.108	0.531 **	0.053	0.497 **		
*F. sp* DNA	0.181	0.016	0.197	0.125	−0.564 ***	0.193	−0.270	−	0.162	0.451 *	−0.241	0.215	0.447 *	
Total DNA	0.124	−0.297	0.565	0.040	−0.107	0.481 **	−0.267	0.201	0.063	0.657 ***	0.343	0.712 ***	0.762 ***	0.515 **

*F.a.*—*F. avenaceum*, *F. c.*—*F. culmorum*, *F. g.*—*F. graminearum*, *F. p.*—*F. poae*, *F. sp.*—*F. sporotrichioides*; coefficients significant at *p* ≤ 0.001 −***; 0.01 − **; 0.05 *.

**Table 8 microorganisms-07-00439-t008:** Coefficients of correlations between seed quality parameters (germination energy—GE, germination capacity—GC, dead seeds, abnormal seedling—AS, fresh ungerminated seeds—FUS) and *Fusarium* head blight index (FHBi), ergosterol (ERG) and type A and B trichothecenes concentrations (Total TCT A, Total TCT B) as well as amount of DNA of five *Fusarium* species in grain samples from organic field.

Variables	GE	GC	Dead Seeds	AS	FUS	FHBi	ERG	Total TCT B	Total TCT B	*F. a.* DNA	*F. c.* DNA	*F. g.* DNA	*F. p.* DNA	*F. sp.* DNA
Final count	0.639 ***													
Dead seeds	−0.511 **	−0.874 ***												
AS	−0.059	−0.327	0.203											
FUS	−0.538 **	−0.579 ***	0.470 **	−0.346										
FHBi	−0.274	−0.533 **	0.456 *	0.360	0.002									
ERG	−0.083	−0.056	0.080	0.257	−0.152	0.036								
Total TCT B	−0.104	−0.219	0.207	0.309	−0.158	0.173	0.182							
Total TCT A	0.097	0.027	0.158	0.056	−0.225	0.315	−0.072	0.162						
*F. a* DNA	−0.023	−0.479 **	0.557 **	0.357	0.117	0.511 **	0.192	−	−					
*F. c* DNA	−0.052	0.068	0.006	−0.202	−0.111	0.077	0.165	0.235	−	−0.210				
*F. g* DNA	−0.127	−0.379 *	0.459 *	0.370 *	−0.003	0.461 **	−0.031	−0.006	−	0.506 **	−0.039			
*F. p* DNA	−0.382 *	−0.577 ***	0.653 ***	0.220	0.282	0.508 **	−0.055	0.208	0.163	0.550 **	0.006	0.546 **		
*F. sp* DNA	−0.338	−0.590 ***	0.593 ***	0.302	0.264	0.584 ***	0.098	−	0.106	0.607 ***	−0.085	0.323	0.530 **	
Total DNA	−0.201	−0.475 **	0.557 ***	0.323	0.089	0.636 ***	0.004	0.216	0.243	0.640 ***	0.214	0.760 ***	0.838 ***	0.623 ***

*F. a.*—*F. avenaceum*, *F. c.*—*F. culmorum*, *F. g.*—*F. graminearum*, *F. p.*—*F. poae*, *F. sp.*—*F. sporotrichioides*; coefficients significant at *P* ≤ 0.001 -***; 0.01—**; 0.05 *.

**Table 9 microorganisms-07-00439-t009:** Average values for groups shown in Figure 3A.

Group	Number of Cultivars	FHBi (%)	DS (%)	ERG (mg kg^−1^)	Total TCT B (μg kg^−1^)	Total TCT A (μg kg^−1^)	*Fusarium* DNA (pg 100 ng^−1^)
1	13	0.4	3.8	2.10	63	4	206
2	5	2.0	8.7	1.48	191	7	610
3	12	0.4	2.8	0.67	43	6	237

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
