# Peer review of "Performance of Winter Wheat Cultivars Grown Organically and Conventionally with Focus on Fusarium Head Blight and Fusarium Trichothecene Toxins"

_microorganisms, 2019, doi:10.3390/microorganisms7100439_

Round 1

Reviewer 1 Report

  1. The manuscript from Goral et al reports about data obtained from one year field trial of 30 winter wheat cvs grown in both organic and conventional conditions.

In my opinion, the main limit of this work is that the trial has been carried out for one year only in one location only and therefore the impact of the environment on the data obtained has not been evaluated. The authors state that "it was decided to provide field experiment...under the same environmental conditions (location, time and weather)"...however without carrying on trials in different locations and/or in different years not easy to obtain robust results. What's your position about this criticism.

Answer 1: Research carried out as part of this work concerns 30 wheat varieties in two systems (conventional and organic) including repetitions in 3 blocks. In this material, comprehensive research was carried out including: phenotypic characters, yellow rust and FHB infection (heading, flowering, plant height, grain yield, yellow rust and FHBi), mineral elements in soil (13 elements) analysis, seed quality (germination energy, germination capacity, abnormal seedlings, dead seedlings and fresh, ungerminated seeds), concentration of DNA of five Fusarium species, analysis of trichothecenes (HT-2 toxin, T-2 toxin, T-2 tetraol, T-2 triol, diacetoxyscirpenol, scirpentriol) , analysis of ergosterol. All these analyzes are very costly overall. For this reason, with the use of extensive statistical analysis, it was decided, based on possible correlations between the examined parameters, to indicate the most relevant for further research. Thus, the results presented are of a pilot nature and further studies are ongoing. However, the results obtained at this stage are so comprehensive and interesting that we considered them worth presenting.

Another criticism is about the organization of the manuscript. Both the title and the introduction lead to suppose that the work was focused on FHB in conventional vs organic grains, however the results and discussion take into account several other aspects, ranging from some soil characteristics to yield and several pathogens etc. In brief, it seems more focused on a comparison of agronomic and qualotative traits of organic vs conventional in general.

Answer 2: Almost all parameters examined and discussed are associated with FHB - they either have an impact or are a consequence of FHB. Therefore, in the title it was decided to express it.

Manuscript was revised to emphasize results for Fusarium head blight, toxins, and Fusarium species. Some results not directly regarding FHB were moved to supplementary files.

3.In fact, the authors carried out a lot of work on the wheat samples, collecting several morpho-physiological and qualitative data.

Answer 3: Thank you for this comment.

  1. Minor observations are:

row 68-69. not clear phrase row 72: ..in both years..Is this correct?? row 90-91: please re-phrase, not clear row 296, Table 3: grain yield (kg/???) row 605: not clear if only ergosterol measures live and dead (DNA?)

Answer 4:  Text was corrected.

Reviewer 2 Report

  1. The introduction section is too long, it should be shortened.

Answer 1: Introduction was shortened and revised.

  1. Verses 121 – 122 „Wheat was grown according to organic farming practices with no chemical disease control and application of fertilizers” - Have been used any other components allowed in organic farming?

Answer 2: The description has been completed

  1. Subchapter 2.4.1 - this section should be edited and described in detail.

Answer 3: DNA isolation technique was described in detail.

4.Authors should include in the manuscript photos of Fusarium infested wheat, as well as photos of Petri dishes of Fusarium isolates.

Answer 4: We included photos of Fusarium infected wheat heads for 4 cultivars. Photos of Fusarium isolates are not available, as we did not isolate Fusarium from grain. Fusarium species were identified and quantified only with real-time PCR.

Round 2

Reviewer 1 Report

The authors' replay are convincing about the fact that the big amount of data compensate the one-year trial and can be considered as the starting point for in deep, future evaluations.